## PROCEEDINGS A

statistical physics, complexity

scaling theory, communicable disease, coronavirus, COVID-19, universality, phase transitions

**Author for correspondence:**
Subir K. Das
e-mail: das@jncasr.ac.in

# A scaling investigation of pattern in the spread of COVID-19: universality in real data and a predictive analytical description

## Subir K. Das

Theoretical Sciences Unit and School of Advanced Materials, Jawaharlal Nehru Centre for Advanced Scientific Research, Jakkur PO, Bangalore 560064, India

SKD, 0000-0001-7414-091X

We analyse the spread of COVID-19, a disease caused by a novel coronavirus, in various countries by proposing a model that exploits the scaling and other important concepts of statistical physics. Quite expectedly, for each of the considered countries, we observe that the spread at early times occurs exponentially fast. We show how the countries can be classified into groups, like *universality classes* in the literature of phase transitions, based on the rates of infections during late times. This method brings a new angle to the understanding of disease spread and is useful in obtaining a country-wise comparative picture of the effectiveness of lockdown-like social measures. Strong similarity, during both natural and lockdown periods, emerges in the spreads within countries having varying geographical locations, climatic conditions, population densities and economic parameters. We derive accurate mathematical forms for the corresponding scaling functions and show how the model can be used as a predictive tool, with instruction even for future waves, and, thus, as a guide for optimizing social measures and medical facilities. The model is expected to be of general relevance in the studies of epidemics.

## 1. Introduction

To minimize damage due to an ongoing epidemic, via lockdown-like physical distancing (PD) measures [1], an

accurate understanding of the pattern in the spread [2–7] of the disease is needed. A popular expectation for the early *natural* spread is [4–9]

$$N = N_0 \exp(mt), \tag{1.1}$$

where $N$ is the number of infections till time $t$, with $N_0$ and $m$ being constants. Deviation from equation (1.1), at later times, with slower spread can occur due to natural reasons as well as because of imposed social restrictions. Interestingly, there exists serious misperception about the complete picture [10].

Advanced methods of analysis [11,12] are needed to obtain an accurate picture of the overall *real* trend. There should be a search for techniques that can help identify the existence of common features in the growth of $N$, in the global scenario, for periods of *actual* natural spread as well as for spread during social restrictions. This objective is in line with the investigation of universality that is observed in phenomena associated with growth during phase transitions in materials [13–27]. Such knowledge can be useful in tracking deficiencies in facilities related to medical testing as well as in identifying inadequacies in social measures.

By applying one such model, here we analyse real data [1] on the spread of COVID-19 [1,9, 28–31], a novel coronavirus disease. This, having *predictive* ability, should find general application in the studies of epidemics. The model is a combination of the scaling theory, more specifically the finite-size scaling model of statistical physics [11,12,32–39], and other advanced techniques [35,40–44] associated with the studies of singularities in phase transitions [13,14,16–27,45].

Like in materials, concepts of phase transitions and universality exist also for phenomena related to the societies [46,47] of living beings. *Ideally* these should apply to the spread of infectious diseases among human populations as well. This is despite differences in cultures and governments. These systems are, in a way, part of the currently popular area concerning biologically 'active matter' [48–52].

However, social systems are made of *finite* numbers of constituents. Because of this and other reasons, as stated above, equation (1.1) describes the spread of a disease only for a short duration, and $N$ remains always finite. Such finiteness exists in the context of materials as well. In reality this is true, say, in nanoscopic systems. Due to technical reasons, this is also encountered in the computer simulations of phase transitions in model systems [11,12,32–39]. This constraint is, however, exploited to an advantage, with the help of the finite-size scaling theory [11,12,32,34,35,39], to arrive at conclusions on universality even for the thermodynamically large systems. Here, of course, in the disease context, finiteness is a fact. We will show, how this theory can be effectively and beneficially used in such situations.

In contrast to some of the recent literature, we observe that in all the considered countries the early growth obeys equation (1.1). We show, how from the shapes of the overall growth curves, the countries, *in fact*, can be grouped into classes, within each of which finite-size-type scaling being satisfied. This aspect, via inter-class comparisons, can alert the administrative authorities on the need for optimization of the PD. We derive a mathematical form for the corresponding scaling picture, that can be useful for influencing the future of the spread. This method is quite effective, providing accurate information by taking certain fluctuations into consideration in a natural way.

## 2. Phase transitions and universality: relevance in disease dynamics

In the vicinity of the critical points of materials, various thermodynamic properties exhibit singular behaviour. For a quantity $X$, one writes [13,14,16–18]

$$X \sim \epsilon^{-x}. \tag{2.1}$$

Here $\epsilon$, say, for a temperature ($T$) driven transition, is the distance of $T$ of a considered thermodynamic state point from that at the critical point and $x$ is a critical exponent. A key point

concerning such phase transitions is the divergence of the correlation length $\xi$:

$$\xi \sim \epsilon^{-\nu}. \tag{2.2}$$

Similar anomalous behaviour is true for certain other thermodynamic quantities as well. Interestingly, the values of the critical exponents are in a strong way independent of the materials, implying universality [13,14,16–18].

Such singularities and universality exist in dynamics as well [14,19,20,24]. In the out-of-equilibrium situations [14,19,20], when a system is quenched to a state point lying inside a miscibility gap, from a homogeneous single phase region, it falls unstable to fluctuations. The system's approach towards the new equilibrium consists of the formation of domains rich in one or the other components or species, and growth of their average size as

$$\ell \sim t^\alpha. \tag{2.3}$$

In this case, the growth exponent $\alpha$ is one of the quantities whose value is universal within a class of systems [20].

It is well known that in the equilibrium context there exist corrections [13,53] to the singularities. These appear in various powers of $\epsilon$ and become important when the state points are far away from the critical point. In the case of non-equilibrium dynamics $t$ is analogous to $1/\epsilon$. In this domain, $t = \infty$ is, thus, *equivalent* to a critical point, at which the characteristic length $\ell$ diverges. In this case also, for $t << \infty$, there are expected to be corrections similar to the equilibrium critical phenomena. The sources of these corrections [35,36] can be the curvature dependence of interfacial tension or that of relevant transport coefficients when the domains are small during a growth process. In the context of disease also corrections can enter through the powers of time.

As in the case of thermodynamic limit behaviour, the concept of universality exists for finite-size nature as well [11,12,32–35]. For an epidemic, these behaviours may correspond to exponential spreads according to equation (1.1) in the beginning, and deviations from this at later times. Thus, the objective, in this area, should be to classify countries based on the patterns of spread in these two regimes. Here countries can be identified with materials as well as finite systems, given that the populations are non-divergent. However, unlike the studies of phase transitions in computers, here the size of a system is not known, if the epidemic is ongoing. This provides the *scope of prediction*. We mention here, as a passing remark, that since the early spread, within a wave, is expected to be exponential, we will use a classification scheme based only on spread at late times.

## 3. The scaling model

## (a) Background

From the existing literature on phase transitions, we first discuss the finite-size scaling method in the equilibrium context [11,12]. From equations (2.1) and (2.2), one has

$$X \sim \xi^{x/\nu}. \tag{3.1}$$

A key point governing the anomalies in phase transitions, as stated above, in both equilibrium and non-equilibrium contexts, is the divergence of a characteristic length [13,14,16–27]. In finite systems such divergences get restricted. For example, $\xi$ cannot grow beyond $L$, the size of the system [11]. However, there is scaling of this length with the system size [11,12,34,35], as the critical point, corresponding to the system, is reached. Thus, at the critical point, where $\xi = L$, one writes the singularity of equation (3.1) as [11,12]

$$X \sim L^{x/\nu}. \tag{3.2}$$

Numerical studies exploit this fact to extract the critical behaviour of a quantity in the large system size limit. Here note that a true critical point exists only when $L = \infty$. The *finite-size* critical points

[12] are purely theoretical concepts. However, following a scaling behaviour, this $L$-dependent quantity tends towards the correct thermodynamic critical point.

Clearly, equation (3.2) displays the expected behaviour when $X$ is estimated at the finite-size critical points [33,54]. This form, for $L = \xi$, is bridged with that for $L \gg \xi$ via the introduction of a scaling function $Y(y)$ as [11,12,32]

$$X = Y(y)L^{x/\nu}. \tag{3.3}$$

Here, $y$ ($= (L/\xi)^{1/\nu}$) is a dimensionless scaling variable, $\xi$ having the thermodynamic limit behaviour as expressed in equation (2.2). Thus, $y$ provides information on the deficiency of the size of a system with respect to the thermodynamic limit. $Y$ is a constant in the $y = 0$, i.e. in the perfect finite-size limit [see equation (3.2)]. For large $y$, one has

$$Y \sim y^{-x}. \tag{3.4}$$

The power-law in equation (3.4) is consistent with the expected divergence of $X$, as in equation (2.1), when $L = \infty$.

In this finite-size scaling method, the correct behaviour of a quantity is identified by observing collapse of data [12], along with the satisfaction of the limiting behaviours, from different system sizes, for $Y$. In the collapse experiments, one treats the values of the exponents as adjustable parameters.

For phase transitions at higher dimensions ($d$), the method is a bit more elaborate [55]. When $d$ is larger than the upper critical dimension, there exists another length scale, competing with the size of the system, and an exponent. In such a situation it becomes necessary to vary the latter as well, to obtain data collapse.

In growth problems concerning the kinetics of phase transitions [20,21], with which the current problem is more closely related, $\ell$ diverges according to equation (2.3). In this case, in the long time limit $\ell = L$ and one writes for the scaling *ansatz* [33–35,39]

$$\ell = Y(y)L, \tag{3.5}$$

with $y = (L/\ell)^{1/\alpha}$ ($= L^{1/\alpha}/t$). Here, in the $y \to \infty$ (i.e. $L \to \infty$ or $t \to 0$) limit [34,35], one should have

$$Y \sim y^{-\alpha}, \tag{3.6}$$

while $Y$ is a constant for $y = 0$. Like equation (3.4), (3.6) is also written by keeping the behaviour of $\ell$ in the thermodynamic system-size limit in mind [see equation (2.3)]. In both the situations, equilibrium and out-of-equilibrium, as also in the case of disease spread, corrections are expected to be present in the finite-size scaling as well, when the system sizes are very small [12].

## (b) Formulation of scaling in the context of disease spread

It is possible to formulate a finite-size scaling model for an epidemic as well, even when the spread is exponentially fast during an early period. There the size of a system should be $N_s$, the population that is infected when the spread has stopped, i.e. the epidemic has died out, or the infected population at the end of a given *wave*. $N_s$ should not necessarily be proportional to the total population of a country, a reason being that the rate of infection and efficiency of control are different in different countries, having no obvious relation with the population size. If the rates of spread are different, questions may be raised on the validity of the method in the context of establishing universality. Note that there may still be uniqueness in the overall functional form, with non-universality only in certain metric factors, such as $m$ in equation (1.1). To obtain collapse of data, scaling of time by $m$ will then be a necessity.

Here, the scaling *ansatz* is

$$\ln n = Y(y) \ln n_s, \tag{3.7}$$

with

$$y = \frac{\ln n_s}{mt}, \tag{3.8}$$

$n = N/N_0$ and $n_s = N_s/N_0$. This transforms the problem to that of a power-law anomaly as in a phase transition. We should have

$$Y \sim y^{-1}, \quad \text{for } y \to \infty, \tag{3.9}$$

and one expects,

$$Y \to \text{constant}, \text{when } y \to 0. \tag{3.10}$$

Quantification of the latter, with an accurate form for the convergence, is essential for prediction. We repeat, if an analysis is performed prior to the death of an epidemic, or more specifically, before the end of a wave within it, even the value of $N_s$ can only be predicted. This is unlike the studies of phase transitions in computers where the size of a system is known.

It is not expected that the finite-size behaviour will be the same for all countries. This will lead to the formation of different classes, providing the possibility of inter-class learning. It is worth mentioning here that for an ongoing epidemic with multiple waves, there will rarely be perfect ending of any wave, unless the epidemic truly dies out.

Coming back to the question on corrections to finite-size scaling, in the disease problem $m$ is a deciding factor on the value of $N_s$, the size of a system. This is by assuming that for each of the countries departure from the early exponential form occurs after a fixed period. Thus, the importance of scaling corrections is related to the value of $m$.

## 4. Results

We present results for six countries, viz., Germany (DE), New Zealand (NZ), Ethiopia (ET), United Kingdom (UK), Spain (ES) and Switzerland (CH). We have analysed data till April 30, 2020. Data beyond this date have been used to verify the accuracy of our predictions. The considered periods contain natural growths as well as growths under PD restrictions. The first confirmed cases in these countries were reported on the following dates [1]—DE: 27 January (Jan); NZ: 28 February (Feb); ET: 13 March; UK: 31 Jan; ES: 31 Jan; CH: 25 Feb; all in the year 2020.

### (a) Preliminary interpretation of data

In figure 1*a*, we show $N$ versus $\tau$ plots for three countries, viz. DE, NZ and ET, on a semi-log scale. The group that these countries will form will be referred to as Class *A*. The unit of $\tau$, the time, is one day, and it is counted from the date on which the first confirmed case in a country was reported [1]. It is clear that $N$ and its rate of increase vary drastically from country to country. Despite visible bending, in the chosen semi-log scale, early period spread may be exponential. However, a clear statement about this from a plotting exercise of this type or a confirmation of it from a 'simple' fitting method is ambiguous. This is because of the complexity of the problem, in addition to the lack of confidence in the choice of regimes, in the presence of PD effects at late times and fluctuations owing to inadequacy in testing. The same sets of data are shown in a double-log scale in figure 1*b*. From there one cannot discard the possibility of power-law growths which in fact are suggested [9]. Thus, more careful analysis by employing an advanced technique is needed.

### (b) Basic strategy

From figure 1, it is also clear that in some of the countries 'instabilities' have set in, i.e. infections within the countries have truly started, rather late. Up to a certain time, the value of which is

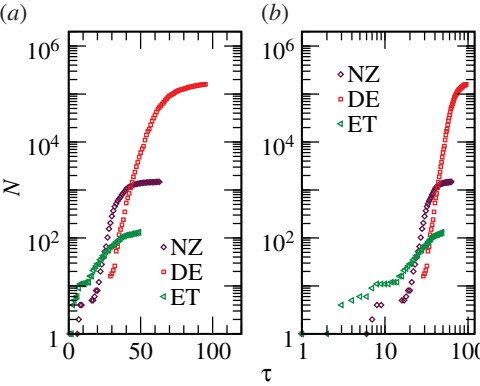

**Figure 1.** (*a*) Plots showing the increase of *N*, in three countries, on a semi-log scale, with time. Here times are counted from the dates on which the first infections were reported. (*b*) Same as (*a*) but here a log-log scale is used. (Online version in colour.)

country specific, the growth that may be present is due to arrivals of patients from abroad. It is, thus, appropriate to start counting time from the day the infections from *within* a country have started getting reported. In other words, for the purpose of analysis of infection rates, judicially chosen onset time ($\tau_0$) should be subtracted from $\tau$ to quantify the growth as a function of $t\ (=\tau-\tau_0)$. We have estimated the value of $\tau_0$, for a given country, as the day from which a steady sharp rise in *N* is visible. Also, we actually examine $n\ (=N/N_0)$. In this way, for each country the growth implies spread by starting from a single patient. This puts all the countries on a fair footing at the beginning. Ideally, $N_0$ should be the value of *N* at $t=0$. But due to early time fluctuation in the data, because of the lack of testing or other deficiencies, we will use it as an adjustable parameter. In fact, it will turn out that the best values of $N_0$, to fair approximations, are the values of *N* at $t=0$.

It is evident from figure 1*a* that if the behaviour is exponential *m* can significantly differ among the countries. The deviations from the early behaviour, exponential or otherwise, say at $N=N_d$, are analogous to the appearance of the finite-size effects [19,35]. In a standard phase transition problem, the characteristic length at the departure of a quantity from the thermodynamic limit behaviour, i.e. the length at the onset of finite-size effects, is proportional to the system size [35,39]. Thus, in the scaling analysis, instead of $N_s$, which is unknown for an ongoing wave within an epidemic, one can work with (country-specific) $N_d$. We treat this also, along with *m* and $N_0$, as an adjustable parameter. Accuracy of the outcomes can be checked from direct *n* versus *t* plots, by scaling the *t* axis by *m*.

## (c) Investigation of scaling in real data

In figure 2*a*, we have presented results from our scaling model. Here, we have shown *Y* as a function of *y*, by including data from the same three countries as in figure 1. The collapse of data appears good, implying the formation of a class or a group. The corresponding numerical values of the adjustable coordinates $(N_0, N_d, m)$ are provided in table 1. There we have also included the numbers for $\tau_0$. The satisfaction of such non-trivial scaling by several countries says that there may be no serious problem with the testing in one country compared to the others. This, in turn, points to *a possibility* that undetected positive cases, mostly from the asymptomatic population, remain a fixed fraction of the total throughout. Of course, the less likely possibility of a modified *m*, for the exponential growth, i.e. time-dependent fractions for undetected positive population, can also not be readily ruled out. The dashed line in this figure is an analytical function, a central result of this work, that will be discussed later. At large *y*, the scaled datasets are consistent with the behaviour in equation (3.9) (see the solid line), implying exponential spread at early time.

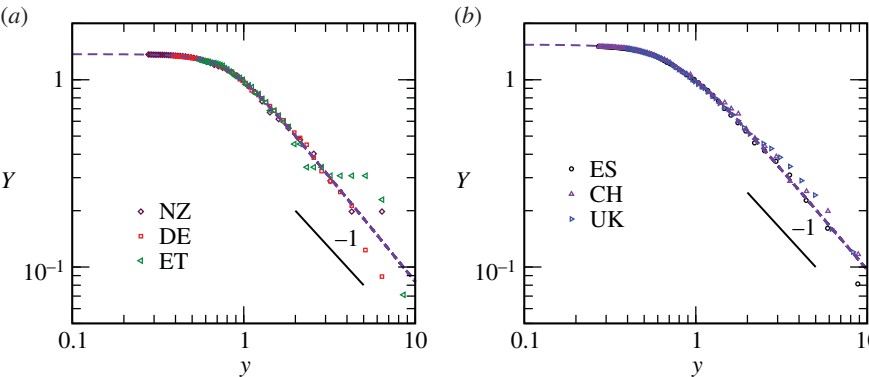

**Figure 2.** (a) Finite-size scaling plots for the spread of COVID-19 in three countries. We have plotted the scaling function $Y$ with the variation of the dimensionless scaling variable $y$. See table 1 for the values of parameters used for the collapse. This group of countries will be referred to as Class $A$. (b) Same as (a) but here we have a different set of countries, to be referred to as Class $B$. The collapse parameters for this class also are listed in table 1. The continuous lines represent equation (3.9). The dashed lines are fits of the analytical form in equation (4.10) to the scaled datasets. The differences in parameter values in the latter equation, particularly in $\theta$, separate the two classes, viz., $A$ and $B$. See text for further details. (Online version in colour.)

**Table 1.** List of certain spreading parameters for various countries. On the left, we have the countries belonging to Class $A$. Information for the countries belonging to Class $B$ are provided on the right side.

| country | DE | NZ | ET | UK | ES | CH |
|---|---|---|---|---|---|---|
| $\tau_0$ | 29 | 17 | 3 | 28 | 26 | 1 |
| $N_0$ | 30 | 3.3 | 5 | 20 | 20 | 4 |
| $N_d$ | 18000 | 290 | 65 | 10500 | 9500 | 1500 |
| $m$ | 0.250 | 0.350 | 0.100 | 0.230 | 0.350 | 0.335 |

Despite disparities in economy, culture, climate and population density, robust collapse *throughout* is quite interesting, even if within a limited set of countries. Given that the value of $m$ is not unique, this observation implies that in the post-exponential regimes, that include periods of PD measures also, these countries are consistently maintaining similar discrepancy from each other as in the pre-crossover regimes. These are interesting facts, requiring attention for understanding, from physicists as well as social scientists. More countries should belong to this class. However, it may not be true that the same set of countries will remain within a class over a very long period. This is because, keeping economic considerations in mind, countries are expected to change policies on PD in a non-uniform manner.

This analysis has the potential of getting data from all the countries overlapped at the very least till the departure from the early behaviour, if this period in different countries are described by a unique function, apart from non-universal [36,37,37] metric factors, which is $m$ for equation (1.1). Such a collapse cannot be obtained via simple scaling of $t$ by $m$, given that the crossover time may not vary much from country to country. Thus, the knowledge of the relative effectiveness of PD in one country compared to the others cannot be accurately captured in such studies. This way our method is helpful in obtaining comparative understanding of the effectiveness of PD by forming Classes.

We have evidence for other groups. An exercise analogous to figure 2a is shown in figure 2b, by using data from UK, ES and CH. Again the collapse is good. This group we will refer to as

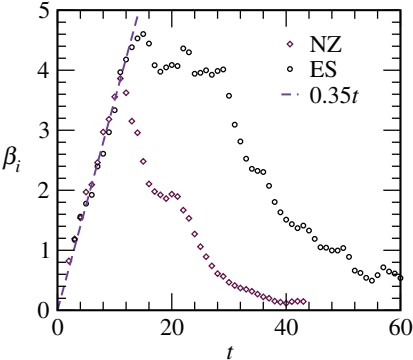

**Figure 3.** The instantaneous exponents (running averaged), $\beta_i$, are plotted versus $t$, for two countries, one each from Class A and Class B. The straight line represents equation (4.2). The used values of $m$, equal ($\simeq 0.35$) for both the countries, were obtained from the data collapse exercise in figure 2—see table 1. (Online version in colour.)

Class B. The adjustable spreading parameters for the countries within this Class are also listed in table 1.

To validate the scaling conclusion on early exponential behaviour, we have calculated $\beta_i$, defined as [35,40–43]

$$\beta_i = \frac{d \ln n}{d \ln t}. \tag{4.1}$$

It is appreciable from equation (4.1) that a purpose of defining such a quantity is to provide information on power-law [40–42], and so, $\beta_i$, in the literature of phase transitions, is referred to as an instantaneous exponent. This quantity, in fact, is helpful in identifying other possibilities as well [43]. It is worth mentioning here, not only in the context of phase transitions, in one or the other form $\beta_i$ has been used in epidemiology also. In figure 3, we have shown $\beta_i$, for two representative countries, one each from Class A and Class B, as a function of $t$. For the behaviour in equation (1.1),

$$\beta_i = mt. \tag{4.2}$$

The early behaviour in the presented datasets is indeed linear, with the *scaling* value(s) of $m$ being quite accurate number(s) for the slope(s). For the latter see table 1. This exercise nicely complements our scaling conclusion on the practically exponential growth at the beginning.

## (d) Construction of a complete analytical description for the scaling function

For prediction, one needs an accurate mathematical expression for $Y(y)$ that must satisfy the limiting expectations noted in equations (3.9) and (3.10). From the analysis thus far it is clear that the large $y$ behaviour is governed by equation (3.9). A reasonable question to ask at this point is: Can the deviation from equation (3.9) be consistently described by an accurate mathematical function? Calculation of another instantaneous exponent,

$$\psi_i = \frac{d \ln Y}{d \ln y}, \tag{4.3}$$

may become useful in answering this. For large $y$, $\psi_i$ is expected to have a flat behaviour, with $\psi_i = -1$. This can be easily checked via equation (3.9). Recall that the early growth is exponential and we have been working with the logarithm of total infections. This calls for a value of unity for $-\psi_i$, when $t$ is small, i.e. $y$ is large. Given that this is a finite-size problem, $\psi_i$ will gradually approach zero in the small $y$ limit. This is related to the fact that in this limit, i.e. when $t \to \infty$, $N$, and so $Y$ [see equation (3.10)], is tending towards a constant. In fact, imposition of such a

**Table 2.** Values of parameters in equation (4.10) for Classes *A* and *B*.

| parameters | $p$ | $\theta$ | $Y_0$ | $b$ |
| --- | --- | --- | --- | --- |
| Class *A* | 1.12 | 3.752 | 1.060 | 0.424 |
| Class *B* | 1.07 | 2.949 | 1.109 | 0.406 |

restriction is necessary, which fortunately is demanded by the finite-size nature of the problem, to produce peaks in the numbers of daily new infections.

Keeping the above-mentioned plateau value as an unknown constant, say, $-p$, rather than promptly using $\psi_i = -1$, one may write, from empirical considerations, that will be verified later,

$$-\frac{1}{\psi_i} = \frac{1}{p} + f(y), \tag{4.4}$$

$f(y)$ being the deciding factor for the decay of $-\psi_i$ to zero. We also want $\psi_i$ to be a monotonic function of $y$. Here, we choose a power-law form for $f(y)$, viz.,

$$f(y) = by^{-\theta}, \tag{4.5}$$

that satisfies the above-mentioned requirements, as further discussed below, when $\theta$ is positive, $b$ being another constant, a positive value for which may also be a necessity. Note that the limiting behaviour

$$\lim_{y \to \infty} f(y) \to 0 \tag{4.6}$$

implies

$$\psi_i = -p, \tag{4.7}$$

which, for $p = 1$, indicates a simple exponential growth, as in equation (1.1), when $t$ is small. On the other hand,

$$\lim_{y \to 0} f(y) \to \infty, \tag{4.8}$$

implying $\psi_i(y = 0) = 0$, corresponds to the convergence of $Y$ to a constant value at late time. Recall that the objective here is to find a compact functional form by combining the early natural behaviour with that during the PD regime.

From equations (4.3), (4.4) and (4.5), one obtains

$$\frac{dY}{Y} = -\frac{dy}{y\left[\frac{1}{p} + by^{-\theta}\right]}. \tag{4.9}$$

After simple algebra, aided by the substitution $b + y^{\theta}/p = z$, one is led to the expression

$$Y(y) = Y_0 \left(b + \frac{y^{\theta}}{p}\right)^{-p/\theta}, \tag{4.10}$$

with $Y_0$, a constant of integration, appearing as an amplitude. Equation (4.10) may apply to the finite-size problems in kinetics of phase transitions as well.

The scaling functions corresponding to the two groups, obtained by fitting the collapsed datasets to equation (4.10), are shown in figures 2*a* and 2*b*, along with the real datasets. The quality of fits certainly appears good. See table 2 for the parameter values. Given that $p$, as expected, appears to be close to unity for all countries, $\theta$ is the quantity that divides the countries into classes. This quantity contains information about how rapidly $\psi_i$ vanishes in the long time limit.

When $y = 0$, one has

$$Y(0) = \frac{Y_0}{b^{p/\theta}}. \tag{4.11}$$

For the quoted parameter values in table 2, $Y(0)$ is larger than unity. This can also be appreciated from figure 2. For typical problems on the kinetics of phase transitions, for which the lengths in

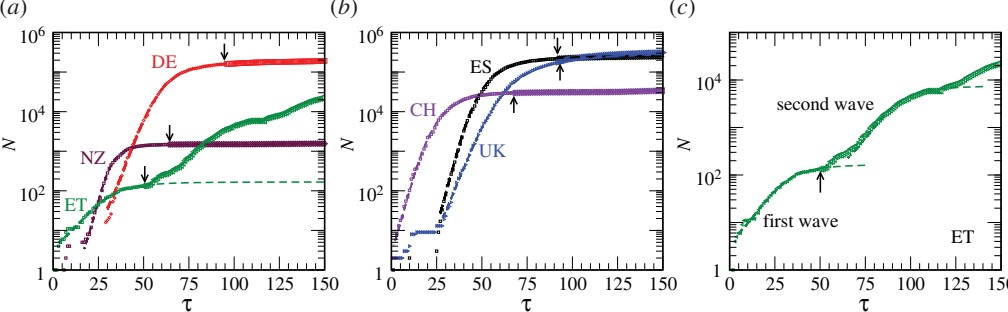

**Figure 4.** (*a*) The dashed lines (mostly indistinguishable from the symbols) represent the spreads, as described by equations (4.10), (4.12) and (4.13), in the countries belonging to Class *A*, on a semi-log scale. The symbols are real data [1]. The data up to 30 April, that are used for obtaining the mathematical description, are represented by smaller symbols. Bigger symbols, beginnings of which are marked by the arrows, are used for the periods beyond April 30. The latter datasets are plotted to verify the accuracy of the predictions. (*b*) Same as (a) but here the countries belong to Class *B*. (c) Here, we have demonstrated that the data for the second wave in ET can be described by the analytical form obtained for the scaling function for the first wave for Class *A*, only after replacing $\tau_0$ and $N_0$ by the new values, viz., 65 and 225. (Online version in colour.)

the equilibrium limits are known, $Y(0) = 1$. The source of this discrepancy lies in the difference between $N_s$ and $N_d$. Recall that since the actual system size $N_s$ is not known, we have been working with $N_d$ here, which is only proportional to the former.

## (e) Demonstration of predictive ability of the model

We observe that the predictions, with the power-law form for $f(y)$, are quite accurate for reasonably long future periods, for all the considered countries. We refer the readers to figure 4 where we have demonstrated this fact for $N$. Exercises related to Class *A* and Class *B* are shown in figure 4*a*,*b*, respectively. There we have presented the real data on $N$ for each of the considered countries by symbols. Data up to April 30, using which the scaling functions were obtained, are represented by smaller symbols. Data from periods beyond this date are represented by bigger symbols. These are practically indistinguishable from the dashed lines, that are obtained, through fitting means, from the analytical function in equation (4.10), for periods even close to three months. A large part of the maximum deviation of up to about 15%, that may be present, might have resulted from manual errors in the estimation of scaling parameters via the collapse experiments. Other reasons for such disagreements can lie in the existence of weak new waves, due to temporary relaxation or non-observance of PD restrictions. These are further discussed below. Note that the growth picture in figure 4 is connected to the scaling function via:

$$N = N_0 \left(\frac{N_d}{N_0}\right)^Y,$$

(4.12)

with time being expressed as

$$\tau = \tau_0 + \frac{1}{my} \ln \frac{N_d}{N_0}.$$

(4.13)

Equations (4.10), (4.12) and (4.13) are the key analytical expressions of this paper.

Given that the data collapse is obtained for the transformed functional form, i.e. the scaling was constructed by taking the logarithm of the number of infections, any minor error in $Y$, due to *manual* inaccuracy in any of $N_0$, $N_d$ and $m$, is expected to magnify the effect significantly in $N$. An error less than 15%, for $N$, over long enough periods, is therefore quite small. Nevertheless, there, of course, exists scope for making improvement in $Y$.

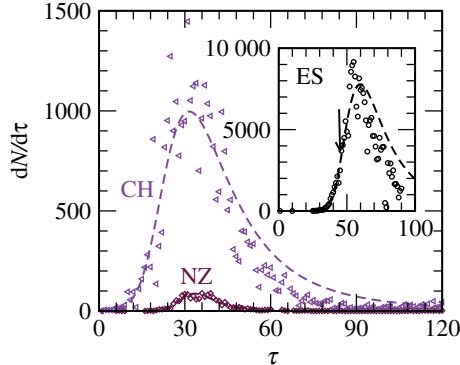

**Figure 5.** Comparisons of the real data (symbols) for daily new cases with the analytical forms (dashed lines) for different countries. Note that the daily new cases are time derivatives of the total numbers presented in figures 1 and 4. Similarly, the dashed lines are time derivatives involving the combination of equations (4.12) and (4.13). Inset shows the case of Spain. In this case, the scaling function was obtained by fitting the data till a date (see the downward arrow) that was significantly before the appearance of the peak. Here, we have used the value of $p$ from table 2. (Online version in colour.)

For ET the data beyond 30 April are not described by the analytical function. Multiple outbreaks, each being of small magnitude though, have been observed in this country till the first half of August. The new waves for ET can be described by the scaling function obtained by using the data sets of the first wave with reasonable accuracy. In figure 4$c$, we have shown data for ET again that include all the waves till 8 August. As in figures 4$a$,$b$, data up to 30 April and beyond are represented, respectively, by small and big symbols. To make the analytical construction for the second wave in ET, we have only changed the values of $\tau_0$ and $N_0$ in the scaling function for the first wave. See the caption of figure 4 for these numbers. It is important to note that both $m$ and the exponential period are same for the first two waves for this country to a good degree of accuracy. Overall, the model is, thus, useful in describing future situations, including fresh outbreaks. With respect to the new waves, however, we do not always expect such strong similarity with the previous waves. Furthermore, for new waves it is more appropriate to analyse data by starting with the number of active cases.

In figure 5, we show the comparison between our analytical form and the real data for numbers of daily new infections. Note that the daily new cases are the time derivatives of $N$. In the main frame, we have included data from two countries, one each from Class $A$ and Class $B$. The agreements are quite satisfactory. Even though the periods of predictions start after the appearances of the peaks, such agreements are non-trivial. For inappropriate functions instabilities are expected to show up. In fact, our model is capable of making accurate prediction even if the scaling function is obtained by using data up to a time that is before the appearance of the peak. This is demonstrated in the inset of this figure for ES. In this case, the analytical plot was obtained by fitting data up to 18 March, a date considerably before the appearance of the actual peak. Here also the shape, location and height of the peak are in good agreement. It is worth mentioning that even if the overall agreement between the analytical functions and the real data in figure 4 appears good, picking up the peaks correctly is non-trivial. It is important to note here that in the standard phase transition problems, for which typically $\alpha \leq 1$, with finite system sizes no such peaks appear in the time derivatives of growth data.

The reason behind the accuracy of the prediction of the model can be understood from figure 6. There we have shown the dependence of $\psi_i$ on $y$. Indeed a power law for $f(y)$ [refer to equations (4.4) and (4.5)] emerges (see the inset). This figure suggests that a steady pattern in the spread sets in rather early. Furthermore, the result indicates again that $p$ is in very good agreement with unity (see the horizontal line in the main frame), i.e. the early spread is essentially exponential. In the ordinate of the inset, we have used the variable $-1/\psi_i - 1$. The subtraction of unity was necessary to obtain accurate information on the exponent $\theta$ from the double-log plot.

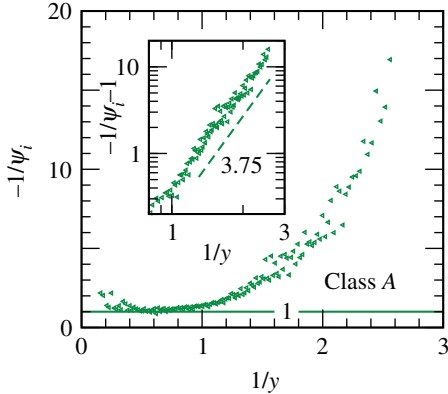

**Figure 6.** Dependence of the instantaneous exponent $\psi_i$ on the scaling variable $y$ is demonstrated for Class *A*, by combining data from all countries within a single set. The dashed line in the inset is a power-law, exponent for which is mentioned. The solid horizontal line in the main frame has ordinate value unity. In the inset, we have subtracted unity from the ordinate variable. This is because of the expectation that $p = 1$, as also seen in the main frame. Without this subtraction, one would derive a wrong idea on the value of $\theta$ from a log-log plot. See table 2 for the value of this exponent for this class. Clearly, the number quoted in the inset is in almost perfect agreement with that. (Online version in colour.)

Note that often due to the presence of a non-zero offset or background, which is $1/p$ here [consult equation (4.4) or equation (4.7)], misleading conclusions on the power-law exponents are arrived at from double-log plots, if data are not available over a substantial range [35].

## (f) A summary of the working steps within the model

At this point, it is worth stating the working steps of the model in brief. The logarithmic conversion of the number of total infections transforms the problem to that of a standard power-law anomaly in the domain of kinetics of phase transitions. Data from different countries are analogous to the results from systems of different sizes, like in the studies of phase transitions in computers with boxes having different volumes or linear dimensions. To establish universality, certain parameters are adjusted (see table 1 for the list, excluding $\tau_0$) to obtain optimum collapse of the transformed data from different countries.

Unlike in computer simulations of phase transitions where the size of a box is known, here the system size is unknown. For the purpose of obtaining scaling collapse it can be treated as an adjustable parameter. In order to predict the number of infections in future, good mathematical forms of the scaling functions must be constructed. For the considered set of countries it appears that the small $y$ behaviour (corresponding to late time) of the scaling function $Y$ has connection with certain power-law. Note that it is this finite-size behaviour that divides the countries into classes, given that for all the studied countries the early growths occur exponentially fast. Once $Y$ is known, from the reverse transformation $N$, as a function of $\tau$, can be obtained.

If obtaining the scaling collapse is not of interest, and the objective is only of a prediction for a particular geographical region, following the logarithmic conversions [see equations (3.7) and (3.8)], after an appropriate identification of $\tau_0$, one can proceed with the fitting to the form in equation (4.10). The input for $Y$, in equation (4.12), however, should not necessarily be that from equation (4.10). For example, for some of the countries that are severely hit by COVID-19, the $\beta_i$ versus $t$ plots tend to plateaus in the post-exponential regime. This fact can be exploited, instead of the behaviour of $\psi_i$, to obtain a dependence of $N$ on $\tau$ in such situations.

## 5. Conclusion

Despite differences in population density, culture, climate and economic parameters, it appears that there is a unique pattern in the spread of COVID-19, including for periods when physical

distancing practices are in place. Our model is capable of figuring out deficiencies in the measures related to PD via the formation of, and comparison among, groups or classes. Even though we have here presented results from two classes with a limited number of countries, we believe that more countries will belong to each of these classes.

The predictive nature of the model we have demonstrated. For this purpose, we have also included an example of a future wave. Accuracy of the predictions, of course, will depend upon consistency of social restrictions over the concerned future period. A crucial point in our prediction is the choice of $f(y)$. For all the countries considered here, power-law behaviour of $f(y)$ appears to describe the data well. It will be interesting to investigate if exponential or logarithmic forms apply to certain groups of countries. In fact, in a less restricted sense, universality can be classified based on the above-mentioned functional forms for $f(y)$.

Our study provides accurate information on the exponential growth in the early natural spread of the epidemic [4,5]. Such growth is commonly attached to an ideal picture for the spread of a rumour, where, say, every person, with the knowledge of it, spreads a hoax to one more individual every next day. In the case of an infectious disease, however, various factors can resist such a spread, even before any strict social measures have been implemented. For example, beyond a certain time either the patients get isolated or get cured or die, thereby leaving the gang of spreaders. Thus, even the natural spread can have crossover to a slower rate. This crossover time should not be much longer than the average incubation period which is said to be about 14 days [1] for COVID-19. However, interestingly, for some countries the crossover times appear longer than even 20 days. A reason behind such an observation can be the continuous arrival of patients from abroad during this period. The role of super-spreaders can also not be ruled out.

It will be interesting to see if our model is capable of identifying the presence of super-spreaders. In network theory, these are like hubs in an assembly of connected nodes [56–58]. With the understanding that a network is an infinite-dimensional object, one certainly works with dimensions higher than the upper critical one. In that case, collapse of data, as previously discussed, requires adjustment of an additional variable [55]. In view of this, it may become necessary to modify the present model. This will, of course, depend upon how the PD measures, that contribute to the finite-size behaviour, influence the overall long-distance network-like character of the disease dynamics. For super-spreaders, if the individuals are associated with, for example, food supply, social measures may not diminish their effects drastically. This, nevertheless, is a complex and debatable matter. In future, by analysing results of computer simulations of relevant disease models, by keeping the provisions of adding and discarding the super-spreaders, we intend to study this aspect.

We note here that it is possible to derive the mathematical form independently, without getting in entirety into the universality related analysis. In this alternative approach, after the analytical form is obtained, parameters like $p$ and $\theta$ for different countries can be estimated so as to form the Classes.

Data accessibility. This article has no additional data.
Authors' contributions. S.K.D. designed the problem, performed the work and wrote the manuscript.
Competing interests. I declare I have no competing interests.
Funding. Partial financial support for this work was received from the Department of Biotechnology, India (grant no. LSRET-JNC/SKD/4539); and Science and Engineering Research Board of Department of Science and Technology, India (grant no. MTR/2019/001585).
Acknowledgements. The author thanks K. Binder, J.V. Sengers and D. Thirumalai for encouraging comments; M. Zannetti and G.C. Das for interesting queries; D. Dhar, R. Ramaswamy and S. Puri for certain information on the literature of epidemiology; and S. Narasimhan for a careful reading of the manuscript.

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
