## [Peer Review File · Proceedings. Mathematical, Physical, and Engineering Sciences]

Review History

RSPA-2020-0689.R0 (Original submission)

Review form: Referee 1

Is the manuscript an original and important contribution to its field?

Good

Is the paper of sufficient general interest?

Good

Is the overall quality of the paper suitable?

Good

Can the paper be shortened without overall detriment to the main message?

Yes

Do you think some of the material would be more appropriate as an electronic appendix?

No

Do you have any ethical concerns with this paper?

No

Recommendation?

Accept with minor revision (please list in comments)

Comments to the Author(s)

In "A Scaling Investigation of Pattern in the Spread of COVID-19: Universality in real data and a predictive analytical description" the author studies a subject with obvious practical interest. The author develops a predictive scaling method on the covid 19 spreading in analogy with phase transition universality class behaviour, by determining critical exponents and the scaling relations related to the spreading phenomena as functions of the time. The manuscript is well written and the obtained results are of interest of a wider audience. I recommend accepting the manuscript for publication after some modifications described in the following:

1) In the Sec. II, the author presents the general behaviour of phase transitions close to a critical point when the correlation length diverges as a power law. When the finite system approaches to the critical point, the correlation length should be in the same order of the system size. This is fundamental for the finite size scaling theory as correctly stated by the author. However, the author may add a comment about what happens with the scaling functions when the phase transition occurs above the upper critical dimension. In fact, the scaling of the steady state observables and the respective data collapses are done as function of the shift exponent (below the upper critical dimension the shift exponent and correlation length exponent are equal). I refer the author to [PRL 98, 258701 (2007)].

2) In fact, an important phenomena regarding the spread of an infection is the presence of superspreaders. In general, a usual method to study spreading phenomena is to consider a subadjacent network in order to describe the human relationships. In a network with connected nodes, a superspreader is recognized as a hub (see, for example, [Networks, 2nd ed., Mark Newman, Rev. Mod. Phys., 80, 1275, and Rev. Mod. Phys., 87, 925]). Networks are "infinite dimensional" objects. Therefore, they are above the upper critical dimension. Regarding the real spreading phenomena, the method devised in the article can help to identify the presence of superspreaders? Superspreaders are mentioned in the conclusions, however, a comment on this subject may be added. In addition, does the author expect any relationship between physical distancing and the presence of superspreaders?

3) In the third paragraph of the Sec. II, the author states the usual scaling ansatz in the long time limit for a spreading process. Please, add a comment on the scaling corrections. Scaling corrections could also be included in the "m" term in the Eq. 1?

4) Is the fitting according to eq. 9 following the parameters on Table II the main difference between the two groups of countries in Fig. 2? Please, include the information in the respective caption.

5) Can we speak about two "universality classes" according to the long time limit as hinted by the names "Class A" and "Class B"? If yes, what is the relevant exponent that selects between the two classes? Is the limit of ψ_i for $y \rightarrow 0$ ($t \rightarrow \infty$)?

6) Minor changes:

a) Please, make the figure captions more self contained, with the exception of Fig 4. Cross-ref the necessary scaling relations and tables in the captions would be welcome;

b) In Table 1, please separate the two groups of countries by adding a new row with merging cells (one cell matching the NE, NZ and ET columns and other cell matching the UK, ES and CH columns;

c) Line 47 of pag. 12: "After some algebra, analogous to phase transitions" is almost cryptic. Please, include a little more detailed explanation.

d) Line 11 of page 16: "unique-ness".

Review form: Referee 2

Is the manuscript an original and important contribution to its field?

Acceptable

Is the paper of sufficient general interest?

Good

Is the overall quality of the paper suitable?

Marginal

Can the paper be shortened without overall detriment to the main message?

Yes

Do you think some of the material would be more appropriate as an electronic appendix?

No

Do you have any ethical concerns with this paper?

No

Recommendation?

Major revision is needed (please make suggestions in comments)

Comments to the Author(s)

Your article contains good ideas, I agree with it. But, please, write it more carefully in a more understandable and organized way. I recommend a revision by a person whose mother tongue is english.

Decision letter (RSPA-2020-0689.R0)

10-Nov-2020

Dear Dr Das

The Editor of Proceedings A has now received comments from referees on the above paper and would like you to revise it in accordance with their suggestions which can be found below (not including confidential reports to the Editor).

Please submit a copy of your revised paper within four weeks - if we do not hear from you within this time then it will be assumed that the paper has been withdrawn. In exceptional circumstances, extensions may be possible if agreed with the Editorial Office in advance.

Please note that it is the editorial policy of Proceedings A to offer authors one round of revision in which to address changes requested by referees. If the revisions are not considered satisfactory by the Editor, then the paper will be rejected, and not considered further for publication by the journal. In the event that the author chooses not to address a referee's comments, and no scientific justification is included in their cover letter for this omission, it is at the discretion of the Editor whether to continue considering the manuscript.

- Acknowledgements
- Funding statement

To revise your manuscript, log into <http://mc.manuscriptcentral.com/prsa> and enter your Author Centre, where you will find your manuscript title listed under "Manuscripts with Decisions." Under "Actions," click on "Create a Revision." Your manuscript number has been appended to denote a revision.

You will be unable to make your revisions on the originally submitted version of the manuscript. Instead, revise your manuscript and upload a new version through your Author Centre.

When submitting your revised manuscript, you will be able to respond to the comments made by the referee(s) and upload a file "Response to Referees" in "Section 6 - File Upload". Please use this to document how you have responded to the comments, and the adjustments you have made. In order to expedite the processing of the revised manuscript, please be as specific as possible in your response to the referee(s).

IMPORTANT: Your original files are available to you when you upload your revised manuscript. Please delete any unnecessary previous files before uploading your revised version.

When revising your paper please ensure that it remains under 28 pages long. In addition, any pages over 20 will be subject to a charge (£150 + VAT (where applicable) per page). Your paper has been ESTIMATED to be 13 pages.

Once again, thank you for submitting your manuscript to Proc. R. Soc. A and I look forward to receiving your revision. If you have any questions at all, please do not hesitate to get in touch.

Yours sincerely

Raminder Shergill
proceedingsa@royalsociety.org

on behalf of
 Professor Vincenzo Capasso
 Board Member
 Proceedings A

Reviewer(s)' Comments to Author:

Referee: 1

Comments to the Author(s)

In "A Scaling Investigation of Pattern in the Spread of COVID-19: Universality in real data and a predictive analytical description" the author studies a subject with obvious practical interest. The author develops a predictive scaling method on the covid 19 spreading in analogy with phase transition universality class behaviour, by determining critical exponents and the scaling relations related to the spreading phenomena as functions of the time. The manuscript is well written and the obtained results are of interest of a wider audience. I recommend accepting the manuscript for publication after some modifications described in the following:

1) In the Sec. II, the author presents the general behaviour of phase transitions close to a critical point when the correlation length diverges as a power law. When the finite system approaches to the critical point, the correlation length should be in the same order of the system size. This is fundamental for the finite size scaling theory as correctly stated by the author. However, the author may add a comment about what happens with the scaling functions when the phase

transition occurs above the upper critical dimension. In fact, the scaling of the steady state observables and the respective data collapses are done as function of the shift exponent (below the upper critical dimension the shift exponent and correlation length exponent are equal). I refer the author to [PRL 98, 258701 (2007)].

2) In fact, an important phenomena regarding the spread of an infection is the presence of superspreaders. In general, a usual method to study spreading phenomena is to consider a subjacent network in order to describe the human relationships. In a network with connected nodes, a superspreader is recognized as a hub (see, for example, [Networks, 2nd ed., Mark Newman, Rev. Mod. Phys., 80, 1275, and Rev. Mod. Phys., 87, 925]). Networks are "infinite dimensional" objects. Therefore, they are above the upper critical dimension. Regarding the real spreading phenomena, the method devised in the article can help to identify the presence of superspreaders? Superspreaders are mentioned in the conclusions, however, a comment on this subject may be added. In addition, does the author expect any relationship between physical distancing and the presence of superspreaders?

3) In the third paragraph of the Sec. II, the author states the usual scaling ansatz in the long time limit for a spreading process. Please, add a comment on the scaling corrections. Scaling corrections could also be included in the "m" term in the Eq. 1?

4) Is the fitting according to eq. 9 following the parameters on Table II the main difference between the two groups of countries in Fig. 2? Please, include the information in the respective caption.

5) Can we speak about two "universality classes" according to the long time limit as hinted by the names "Class A" and "Class B"? If yes, what is the relevant exponent that selects between the two classes? Is the limit of ψ_i for $y > 0$ ($t \rightarrow \infty$)?

6) Minor changes:

a) Please, make the figure captions more self contained, with the exception of Fig 4. Cross-ref the necessary scaling relations and tables in the captions would be welcome;

b) In Table 1, please separate the two groups of countries by adding a new row with merging cells (one cell matching the NE, NZ and ET columns and other cell matching the UK, ES and CH columns);

c) Line 47 of pag. 12: "After some algebra, analogous to phase transitions" is almost cryptic. Please, include a little more detailed explanation.

d) Line 11 of page 16: "unique-ness".

Referee: 2

Comments to the Author(s)

Your article contains good ideas, I agree with it. But, please, write it more carefully in a more understandable and organized way. I recommend a revision by a person whose mother tongue is english.

Board Member:

Comments to Author(s):

The authors are welcome to submit a revised version of their manuscript in a substantial way, by taking in due account the comments by both reviewers.

Please address all comments point by point in a response letter.

Author's Response to Decision Letter for (RSPA-2020-0689.R0)

See Appendix A.

RSPA-2020-0689.R1 (Revision)

Review form: Referee 1

Is the manuscript an original and important contribution to its field?

Excellent

Is the paper of sufficient general interest?

Excellent

Is the overall quality of the paper suitable?

Excellent

Can the paper be shortened without overall detriment to the main message?

Yes

Do you think some of the material would be more appropriate as an electronic appendix?

Yes

Do you have any ethical concerns with this paper?

No

Recommendation?

Accept as is

Comments to the Author(s)

I am satisfied with the author responses.

Review form: Referee 2

Is the manuscript an original and important contribution to its field?

Excellent

Is the paper of sufficient general interest?

Good

Is the overall quality of the paper suitable?

Good

Can the paper be shortened without overall detriment to the main message?

Yes

Do you think some of the material would be more appropriate as an electronic appendix?

No

Do you have any ethical concerns with this paper?

No

Recommendation?

Accept as is

Comments to the Author(s)

As I said in my previous comment, I believe that theory of phase transitions can be successfully applied to a wide range of phenomena and situations. I encourage you to follow this line.

Decision letter (RSPA-2020-0689.R1)

05-Jan-2021

Dear Dr Das

I am pleased to inform you that your manuscript entitled "A Scaling Investigation of Pattern in the Spread of COVID-19: Universality in real data and a predictive analytical description" has been accepted in its final form for publication in Proceedings A.

Our Production Office will be in contact with you in due course. You can expect to receive a proof of your article soon. Please contact the office to let us know if you are likely to be away from e-mail in the near future. If you do not notify us and comments are not received within 5 days of sending the proof, we may publish the paper as it stands.

COVID-19 rapid publication process: We are taking steps to expedite the publication of research relevant to the pandemic. If you wish, you can opt to have your paper published as soon as it is ready, rather than waiting for it to be published on the scheduled Wednesday.

This means your paper will not be included in the weekly media round-up which the Society sends to journalists ahead of publication. However, it will appear in the COVID-19 Publishing Collection which journalists will be directed to each week

(<https://royalsocietypublishing.org/topic/special-collections/novel-coronavirus-outbreak>)

If you wish to have your paper published immediately please notify proca_proofs@royalsociety.org and press@royalsociety.org

The Royal Society has signed a Wellcome statement on the subject of research findings and data relevant to the coronavirus (COVID-19) outbreak. We are one of several signatories to this statement and our collective aim is to ensure that the relevant research and data are shared rapidly and openly in order to inform the worldwide public health response and to help save lives. We are therefore making papers related to COVID-19 open access free of charge.

Under the terms of our licence to publish you may post the author generated postprint (ie. your accepted version not the final typeset version) of your manuscript at any time and this can be made freely available. Postprints can be deposited on a personal or institutional website, or a recognised server/repository. Please note however, that the reporting of postprints is subject to a media embargo, and that the status the manuscript should be made clear. Upon publication of the definitive version on the publisher's site, full details and a link should be added.

You can cite the article in advance of publication using its DOI. The DOI will take the form: 10.1098/rspa.XXXX.YYYY, where XXXX and YYYY are the last 8 digits of your manuscript number (eg. if your manuscript number is RSPA-2017-1234 the DOI would be 10.1098/rspa.2017.1234).

For tips on promoting your accepted paper see our blog post:
<https://royalsociety.org/blog/2020/07/promoting-your-latest-paper-and-tracking-your-results/>

On behalf of the Editor of Proceedings A, we look forward to your continued contributions to the Journal.

Sincerely,
Raminder Shergill
proceedingsa@royalsociety.org

Reviewer(s)' Comments to Author:

Referee: 2

Comments to the Author(s)

As I said in my previous comment, I believe that theory of phase transitions can be successfully applied to a wide range of phenomena and situations. I encourage you to follow this line.

Referee: 1

Comments to the Author(s)

I am satisfied with the author responses.

Appendix A

This document contains response letters to both the Referees, in addition to a list of changes.

RSPA-2020-0689: Response to the report from Referee 1

I am very happy to read that the Referee finds the work of practical importance, manuscript well written, and the results of broad interest. I am thankful to him/her for recommending publication. In my opinion all the comments are very thoughtful. I have made modifications in line with each of these. Before providing the detailed response I briefly state about the general structural changes that the manuscript has gone through. This will be helpful in identifying the locations where the corrections in response to this Referee's comments have been inserted.

Following the comments from this and the other Referee I have made rearrangements in the presentation with more detailed and systematic discussions and explanations of various issues. For this purpose a new section has been added, viz., Section II. As a result, the old section numbers got uniformly shifted, from Section II onward. Furthermore, I have divided Section IV, that contains the Results, into several subsections. Towards the end of this section I have added new text, by summarizing the overall method. This comes under a separate subsection.

In the new section, viz., Section II, we have placed the description of phase transitions, along with corrections, and discussion on relevance of the topic in the disease context. Here and in some other places, I have put several mathematical expressions, that previously appeared inside the text, as equations. This is by thinking that the readers can use these as quick references for understanding of materials that are presented later. Thus, the old equation numbers also got changed in many places. While making such modifications, I needed to remove certain existing texts from various places, to avoid repetition, without losing continuity or smooth flow of presentation.

Details of response and modifications in connection with the comments from this Referee are provided below. The comments are typed in *italic* and the modifications in response to these appear in **bold**.

1. *In the Sec. II, the author presents the general behaviour of phase transitions close to a critical point when the correlation length diverges as a power law. When the finite system approaches to the critical point, the correlation length should be in the same order of the system size. This is fundamental for the finite size scaling theory as correctly stated by the author. However, the author may add a comment about what happens with the scaling functions when the phase transition occurs above the upper critical dimension. In fact, the scaling of the steady state observables and the respective data collapses are done as function of the shift exponent (below the upper critical dimension the shift exponent and correlation length exponent are equal). I refer the author to [PRL 98, 258701 (2007)].*

I am thankful to the Referee for raising this important point. This is now discussed in Section II (new number III). This section has been divided into two subsections. The added text, in response to this comment, that appears in the first subsection, reads

“For phase transitions at higher dimensions (d) the method is a bit more elaborate [55]. When d is larger than the upper critical dimension, there exists another length scale, competing with the size of the system, and an exponent. In such a situation it becomes necessary to vary the latter as well, to obtain data collapse.”

Here I have cited the Reference

“Phys. Rev. Lett. 98, 258701 (2007)”

that the Referee quoted.

2. In fact, an important phenomena regarding the spread of an infection is the presence of superspreaders. In general, a usual method to study spreading phenomena is to consider a subjacent network in order to describe the human relationships. In a network with connected nodes, a superspreader is recognized as a hub (see, for example, [Networks, 2nd ed., Mark Newman, Rev. Mod. Phys., 80, 1275, and Rev. Mod. Phys., 87, 925]). Networks are "infinite dimensional" objects. Therefore, they are above the upper critical dimension. Regarding the real spreading phenomena, the method devised in the article can help to identify the presence of superspreaders? Superspreaders are mentioned in the conclusions, however, a comment on this subject may be added. In addition, does the author expect any relationship between physical distancing and the presence of superspreaders?

This is another important comment, having connection with the previous one. In response to this I have added in the Conclusion section

“It will be interesting to see if our model is capable of identifying the presence of super-spreaders. In network theory these are like hubs in an assembly of connected nodes [56-58]. With the understanding that a network is an infinite-dimensional object, one certainly works with dimensions higher than the upper critical one. In that case, collapse of data, as previously discussed, requires adjustment of an additional variable [55]. In view of this, it may become necessary to modify the present model. This will, of course, depend upon how the PD measures, that contribute to the finite-size behavior, influence the overall long-distance network-like character of the disease dynamics. For super-spreaders, if the individuals are associated with, for example, food supply, social measures may not diminish their effects drastically. This, nevertheless, is a complex and debatable matter. In future, by analyzing results of computer simulations of relevant disease models, by keeping the provisions of adding and discarding the super-spreaders, we intend to study this aspect.”

There I have cited all the three References mentioned by the Referee on network theory. On the point of identifying the presence of superspreaders, experience needs to be gained via analysis of data coming from simulations of models within which super-spreaders can be added and deleted at will. Based on the knowledge of the observed differences, between studies with and without these entities, from such set-ups, the method can be applied to real data. This, I believe, can be a research problem of serious future interest.

3. In the third paragraph of the Sec. II, the author states the usual scaling ansatz in the

long time limit for a spreading process. Please, add a comment on the scaling corrections. Scaling corrections could also be included in the "m" term in the Eq. 1?

In the revised version of the manuscript I have provided discussions on corrections. This is done in the general phase transition context, for both equilibrium and nonequilibrium problems, as well as in the disease scenario. In my understanding the Referee has asked for a discussion only in connection with the topic of finite-size scaling. However, I have added discussion for the circumstances of thermodynamically large systems as well.

I completely agree with the referee that the finite-size scaling corrections in the disease problem can come through m . In standard phase transition problems corrections are expected for small system sizes. In the disease problems m is a deciding factor on the value of N_s , the size of a system. If a crossover from the early exponential growth occurs, for all the countries within a class, after a fixed period, this indeed is the case. In response to this comment we have added the following texts. For each of the additions appropriate **References have been cited.**

In Section II, it has been written:

“It is well known that in the equilibrium context there exist corrections [13,53] to the singularities. These appear in various powers of ϵ and become important when the state points are far away from the critical point. In the case of nonequilibrium dynamics t is analogous to $1/\epsilon$. In this domain, $t = \infty$ is, thus, equivalent to a critical point, at which the characteristic length ℓ diverges. In this case also, for $t \ll \infty$, there are expected to be corrections similar to the equilibrium critical phenomena. The sources of these corrections [35,36] can be the curvature dependence of interfacial tension or of relevant transport coefficients when the domains are small during a growth process. In the context of disease also corrections can enter through the powers of time.”

The additions in Section III read:

“In both the situations, equilibrium and out-of-equilibrium, as also in the case of disease spread, corrections are expected to be present in the finite-size scaling as well, when the system sizes are very small [12].”

and

“Coming back to the question on corrections to scaling, in the disease problem m is a deciding factor on the value of N_s , the size of a system. This is by assuming that for each of the countries departure from the early exponential form occurs after a fixed period. Thus, the importance of scaling corrections is related to the value of m .”

There I have cited the new Reference

“P.H. Lundow and I.A. Campbell, Physica A 511 40 (2018).”

4. *Is the fitting according to eq. (9) following the parameters on Table II the main difference between the two groups of countries in Fig. 2? Please, include the information in the respective caption.*

Yes, the Referee is correct. According to his/her advice I have included the relevant information in the caption of Fig. 2 that reads [note that Eq. (9) has now become Eq. (24)]

“The dashed lines are fits of the analytical form in Eq. (24) to the scaled data sets. The differences in parameter values in the latter equation, particularly in θ , define the two classes, viz., A and B.”

I have stressed upon this point in the text also, as mentioned below in response to other comments.

5. *Can we speak about two "universality classes" according to the long time limit as hinted by the names "Class A" and "Class B"? If yes, what is the relevant exponent that selects between the two classes? Is the limit of ψ_i for $y \rightarrow 0$ ($t \rightarrow \infty$)?*

Yes, the universality classes are decided by the long time behavior. Essentially, the exponent θ decides on the classes and, as the Referee says, this is related to the behavior of ψ_i in the small y limit. In response to this comment I have added the following texts in the revised manuscript:

In Section II it is stated

“We mention here, as a passing remark, that since the early spread is expected to be exponential, we will use a classification scheme based only on spread at late times.”

In Section IV.D we have mentioned

“Given that p , as expected, appears to be close to unity for all countries, θ is the quantity that divides the countries into classes. This quantity contains information about how rapidly ψ_i vanishes in the long time limit.”

In the abstract also I have briefly mentioned about it.

6. (a) *Please, make the figure captions more self contained, with the exception of Fig 4. Cross-ref the necessary scaling relations and tables in the captions would be welcome.*

Following this advice from the Referee, I have incorporated more information in each of the captions. This is particularly true for **“Figs. 2, 3, 5 and 6”**. Wherever necessary cross-references are also incorporated.

(b) *In Table 1, please separate the two groups of countries by adding a new row with merging cells (one cell matching the DE, NZ and ET columns and other cell matching the UK, ES and CH columns).*

As instructed by the Referee, I now have made the division between the two classes clear in the table. This is also mentioned in the corresponding caption. This reads

“The thin empty column separates the countries belonging to Class A and Class B. On the left we have the countries belonging to Class A. Information for the countries belonging to Class B are provided on the right side.”

(c) *Line 47 of page 12: "After some algebra, analogous to phase transitions" is almost cryptic. Please, include a little more detailed explanation.*

According to the Referee's suggestion, I have provided further details. Several new equations have been inserted, along with texts explaining these as well as overall procedure. Please read the modified presentation that comes under subsection D in Section IV. I am quoting the relevant parts below.

The text below Eq. (17) now reads

“For large y , ψ_i is expected to have a flat behavior, with $\psi_i = -1$. This can be easily checked via Eq. (13). Recall that the early growth is exponential and we have been working with the logarithm of total infections. This calls for a value of unity for $-\psi_i$, when t is small, i.e., y is large. Given that this is a finite-size problem, ψ_i will gradually approach zero in the small y limit. This is related to the fact that in this limit, i.e., when $t \rightarrow \infty$, N , and so Y , is tending towards a constant [see Eq. (14)]. In fact, imposition of such a restriction is necessary, which fortunately is demanded by the finite-size nature of the problem, to produce peaks in the numbers of daily new infections.”

Material after Eq. (18) now reads

“We also want ψ_i to be a monotonic function of y . Here we choose a power-law form for $f(y)$, viz.,

$$f(y) = by^{-\theta}, \quad (19)$$

that satisfies the above mentioned requirements, as further discussed below, when θ is positive, b being another constant, a positive value for which may also be a necessity. Note that the limiting behavior

$$\lim_{y \rightarrow \infty} f(y) \rightarrow 0 \quad (20)$$

implies

$$\psi_i = -p, \quad (21)$$

which, for $p = 1$, indicates a simple exponential growth, as in Eq. (1), when t is small. On the other hand,

$$\lim_{y \rightarrow 0} f(y) \rightarrow \infty, \quad (22)$$

implying $\psi_i(y = 0) = 0$, corresponds to the convergence of Y to a constant value at late time. Recall that the objective here is to find a compact functional form by combining the early natural behavior with that during the PD regime.

From Eqs. (17), (18) and (19), one obtains

$$\frac{dY}{Y} = -\frac{dy}{y\left[\frac{1}{p} + by^{-\theta}\right]}. \quad (23)$$

After some algebra, aided by the substitution $b + y^\theta/p = z$, one is led to the expression

$$Y(y) = Y_0 \left(b + \frac{y^\theta}{p} \right)^{-p/\theta}, \quad (24)$$

with Y_0 , a constant of integration, appearing as an amplitude. Eq. (24) may apply to the finite-size problems in kinetics of phase transitions as well.”

At the end of this subsection I have added:

“When $y = 0$, one has

$$Y(0) = \frac{Y_0}{b^p/\theta}. \quad (25)$$

For the quoted parameter values in Table II, $Y(0)$ is larger than unity. This can also be appreciated from Fig. 2. For typical problems on the kinetics of phase transitions, for which the lengths at the equilibrium limits are known, $Y(0) = 1$. The source of this discrepancy lies in the difference between N_s and N_d . Recall that since the actual system size N_s is not known, we have been working with N_d , which is only proportional to the former.”

(d) *Line 11 of page 16: "unique-ness".*

This word in the revised version has been replaced by **“unique behavior”**.

I believe to have responded to the Referee’s comments satisfactorily and modified the manuscript appropriately. Once again, I am thankful to him/her for insightful remarks. Needless to say, these have helped raise the standard of the article. I hope that the paper will now be accepted for publication.

RSPA-2020-0689: Response to the report from Referee 2

I am happy to see that the Referee finds the ideas good. *As advised by him/her I have now presented and discussed the results in a much more carefully organized way that can provide easier understanding. The manuscript was also read by a colleague whose mother tongue is English.* This has been appropriately acknowledged in the revised version. The details are provided below.

In response to the comments from the Referee, we have incorporated a new section, appearing as Section II. There we have provided a clearer discussion on phase transitions, and drawn connection of it with the spread of disease, in the finite-size scenario, in a more systematic and transparent way. This has been achieved by elaborating on the materials that have been moved to this Section from Sections I and III. This new section also contains additional discussions on corrections and other scaling aspects that are placed in carefully chosen locations, following certain comments from the other Referee. Here and in the next two sections, where we discuss, respectively, the finite-size scaling model and the results, several mathematical expressions are now brought out from the text and put as equations. The readers can use them as ready references while looking at the other parts.

The section on the “Scaling Model” is now divided into two subsections. The first part provides a clearer background in the established context of phase transitions in condensed matter systems. In the second part formulation of the finite-size scaling model, in the disease context, is discussed with an improved clarity.

On the matter of better organization, in addition to the above mentioned implementations, Section IV, that contains the Results, has also now been divided into several subsections. This way the readers will get proper feeling about the locations of materials of different types while moving back and forth in the process of reading. In each of these parts, there exists superior discussion now. The derivation of the scaling function has been put under the subsection “Construction of a Complete Analytical Description for the Scaling Function”. There the clarity has been made better by adding certain intermediate mathematical steps and further discussions. At the end of this section, new text has been added, that summarizes the basic steps under the model. This new text comes under a separate subsection. I believe, this addition will be helpful, particularly for those readers who want quick implementation for practical purposes.

Because of the new additions, to avoid unnecessary repetition, some of the previously appearing texts have been judiciously removed from various places.

The Introduction also appears more precise now, at the same time being clearer.

Accurate details on the major changes are provided below. The modifications are typed in **bold**.

1. In the Introduction the fifth paragraph is substantially new. This reads

“However, social systems are made of *finite* numbers of constituents. Because of this and other reasons, as stated above, Eq. (1) describes the spread of a disease only for a short duration, and N remains finite. Such finiteness exists

in the context of materials as well. In reality this is true, say, in nanoscopic systems. Due to technical reasons, this is also encountered in the computer simulations of phase transitions in model systems [11,12,31-38]. This constraint is, however, exploited to an advantage, with the help of the finite-size scaling theory [11,12,31,33,34,38], to arrive at conclusions on universality even for the thermodynamically large systems. Here, of course, in the disease context, finiteness is a fact. We will show, how this theory can be effectively and beneficially used in such situations.”

Paragraph five of the previous version has been moved to the next section.

2. Section II is newly introduced. There the last paragraph is from Section I in a modified form. The first two paragraphs are expanded forms of materials that previously appeared in the Section with the heading “The Scaling Model”. These now read

“In the vicinity of the critical points of materials, various thermodynamic properties exhibit singular behavior. For a quantity X , one writes [13-17]

$$X \sim \epsilon^{-x}. \quad (2)$$

Here ϵ , say, for a temperature (T) driven transition, is the distance of T of a considered thermodynamic state point from that at the critical point and x is a critical exponent. A key point concerning such phase transitions is the divergence of the correlation length ξ :

$$\xi \sim \epsilon^{-\nu}. \quad (3)$$

Such anomalous behavior is true for certain other thermodynamic quantities as well. Interestingly, the values of the critical exponents are in a strong way independent of the materials, implying universality [13-17].

Such singularities and universality exist in dynamics as well [14,18,19,23]. In the out-of-equilibrium situations [14,18,19], when a system is quenched to a state point lying inside a miscibility gap, from a homogeneous single phase region, it falls unstable to fluctuations. The system’s approach towards the new equilibrium consists of the formation of domains rich in one or the other components or species, and growth of their average size as

$$\ell \sim t^\alpha. \quad (4)$$

In this case the growth exponent α is one of the quantities whose value is universal within a class of systems [19].”

Previously the equations with numbers (2), (3), and (4) appeared in the text.

The third paragraph and addition at the end of the last paragraph in this subsection are in response to a comment from the other Referee.

3. Section III (previous number II), with the heading “The Scaling Model”, has been broken into two subsections. The first one provides the background. Here the basics of finite-size scaling have been discussed in the context of phase transitions in condensed matter systems, for both equilibrium and nonequilibrium phenomena. The first paragraph under this subsection is an expansion of the previously existing text. This reads

“From the existing literature on phase transitions, we first discuss the finite-size scaling method in the equilibrium context [11,12]. From Eqs. (2) and (3) one has

$$X \sim \xi^{x/\nu}. \quad (5)$$

A key point governing the anomalies in phase transitions, as stated above, in both equilibrium and nonequilibrium contexts, is the divergence of a characteristic length [13-26]. In finite systems such divergences get restricted. E.g., ξ cannot grow beyond L , the size of the system [11]. However, there is scaling of this length with the system size [11,12,33,34], as the critical point, corresponding to the system, is reached. Thus, at the critical point, where $\xi = L$, one writes the singularity of Eq. (5) as [11,12]

$$X \sim L^{x/\nu}. \quad (6)$$

Numerical studies exploit this fact to extract the critical behavior of a quantity in the large system size limit. Here note that a true critical point exists only when $L = \infty$. The *finite-size* critical points [12] are purely theoretical concepts. However, following a scaling behavior, this L -dependent quantity tends towards the correct thermodynamic critical point.”

Here as well as in the rest of the subsection a few mathematical expressions have been brought out of the text and put as equations. Furthermore, this subsection contains a few additions in response to the comments from the other Referee. The formulation of the scaling model in the disease context has been put under the second subsection. There also two mathematical expressions have been displayed as equations. These appeared inside the text previously. The last paragraph under this subsection is again an addition in Response to the other Referee’s comment.

4. Section IV (previously III) has been divided into six subsections. In several places I have added minor or major discussions for better clarity. Most significant modifications have been made in subsection D where I have presented the derivation of the scaling function. Please see the following text that appears below Eq. (17)

“For large y , ψ_i is expected to have a flat behavior, with $\psi_i = -1$. This can be easily checked via Eq. (13). Recall that the early growth is exponential and we have been working with the logarithm of total infections. This calls for a value of unity for $-\psi_i$, when t is small, i.e., y is large. Given that this is a finite-size problem, ψ_i will gradually approach zero in the small y limit. This is related to the fact that in this limit, i.e., when $t \rightarrow \infty$, N , and so Y , is tending towards a constant [see Eq. (14)]. In fact, imposition of such a restriction is necessary, which fortunately is demanded by the finite-size nature of the problem, to produce peaks in the numbers of daily new infections.”

and material that appear below Eq. (18)

“We also want ψ_i to be a monotonic function of y . Here we choose a power-law form for $f(y)$, viz.,

$$f(y) = by^{-\theta}, \quad (19)$$

that satisfies the above mentioned requirements, as further discussed below, when θ is positive, b being another constant, a positive value for which may also be a necessity. Note that the limiting behavior

$$\lim_{y \rightarrow \infty} f(y) \rightarrow 0 \quad (20)$$

implies

$$\psi_i = -p, \quad (21)$$

which, for $p = 1$, indicates a simple exponential growth, as in Eq. (1), when t is small. On the other hand,

$$\lim_{y \rightarrow 0} f(y) \rightarrow \infty, \quad (22)$$

implying $\psi_i(y = 0) = 0$, corresponds to the convergence of Y to a constant value at late time. Recall that the objective here is to find a compact functional form by combining the early natural behavior with that during the PD regime.

From Eqs. (17), (18) and (19), one obtains

$$\frac{dY}{Y} = -\frac{dy}{y[\frac{1}{p} + by^{-\theta}]}. \quad (23)$$

After some algebra, aided by the substitution $b + y^\theta/p = z$, one is led to the expression

$$Y(y) = Y_0 \left(b + \frac{y^\theta}{p} \right)^{-p/\theta}, \quad (24)$$

with Y_0 , a constant of integration, appearing as an amplitude. Eq. (24) may apply to the finite-size problems in kinetics of phase transitions as well.”

as well as the last paragraph under this subsection:

“When $y = 0$, one has

$$Y(0) = \frac{Y_0}{b^{p/\theta}}. \quad (25)$$

For the quoted parameter values in Table II, $Y(0)$ is larger than unity. This can also be appreciated from Fig. 2. For typical problems of phase transitions, for which the lengths at the equilibrium limits are known, $Y(0) = 1$. The source of this discrepancy lies in the difference between N_s and N_d . Recall that since the actual system size N_s is not known, we have been working with N_d , which is only proportional to the former.”

In the second part of last paragraph under subsection E we have added new text that provides explanation on the data presented in the inset of Fig. 6. This reads

“In the ordinate of the inset we have used the variable $-1/\psi_i - 1$. The subtraction of unity was necessary to obtain accurate information on the exponent θ from the double-log plot. Note that often due to the presence of a nonzero offset or background, which is $1/p$ here [consult Eq. (18) or Eq. (21)], misleading conclusions on the power-law exponents are arrived at from double-log plots, if data are not available over a substantial range.”

The last subsection is new. Here we have written

“At this point it is worth stating the working steps of the model in brief. The logarithmic conversion of the number of total infections transforms the problem to that of a standard power-law anomaly in the domain of kinetics of phase transitions. Data from different countries are analogous to the results from systems of different sizes, like in the studies of phase transitions in computers with boxes having different volumes or linear dimensions. To establish universality, certain parameters are adjusted (see Table I for the list, excluding τ_0) to obtain optimum collapse of the transformed data from different countries.

Unlike in computer simulations of phase transitions where the size of a box is known, here the system size is unknown. For the purpose of obtaining scaling collapse it can be treated as an adjustable parameter. In order to predict the number of infections in future, good mathematical forms of the scaling functions must be constructed. For the considered set of countries it appears that the small y behavior (corresponding to late time) of the scaling function Y is a power-law. Note that it is this finite-size behavior that divides the countries into classes, given that for all the studied countries the early growths occur exponentially fast. Once Y is known, from the reverse transformation N , as a function of τ , can be obtained.

If obtaining the scaling collapse is not of interest, and the objective is only of a prediction for a particular geographical region, following the logarithmic conversions [see Eqs. (11) and (12)], after an appropriate identification of τ_0 , one can proceed with the fitting to the form in Eq. (24). The input for Y , in Eq. (25), however, should not necessarily be that from Eq. (24). E.g., for some of the countries that are severely hit by COVID-19, the β_i versus t plots tend to plateaus in the post-exponential regime. This fact can be exploited, instead of the behavior of ψ_i , to obtain a dependence of N on τ in such situations.”

5. Minor modifications have been made in the last section as well, in Response to the comments from this Referee.

6. We have incorporated several other clarifications/modification. Below I mention a few.

(a) Clear mention about the adjustable parameters in the data collapse experiments of Fig. 2.

(b) Precise statement on the classification scheme.

(c) Better discussion on sources of errors.

(d) Possible future direction on identification of super-spreaders.

Etc.

Captions of all the Figures and Tables have also been made more informative.

Once again, I am thankful to the referee for constructive criticism. I believe that the presentation now is much better and hope that the paper will now be accepted for publication. I am indeed happy that the Referee agrees with the ideas and finds them good.

RSPA-2020-0689: List of Changes (for Editorial/Journal Office)

1. The second sentence in the Abstract is new.
2. The fifth paragraph from the Introduction has been moved to the end of Section II, after modifications in a few places. The current fifth paragraph under this section is substantially new.
3. Section II is new. Materials here are brought from Sections I and III (previously II). After this reorganization the discussions in this section have been expanded.
4. Section III (previously II) has been divided into two subsections. The first and the last paragraphs under the first subsection contain substantially modified text. The fourth paragraph in this subsection and the last paragraph under the second subsection are new.
5. Section IV (previously III) has been divided into six subsections. The text under the last subsection is entirely new. In the fourth subsection materials below Eq. (17) have undergone significant modifications, including additions of a few new equations. In the fifth subsection, the second half of the last paragraph is new that starts with “In the ordinate of the inset”.
6. In Section V (previously IV) fourth paragraph is new.
7. All the Figure captions have been modified, major additions being in those for Figs. 2, 3, 5 and 6. We have added new information in the caption of Table I as well.
8. We have added ‘Acknowledgements’.
9. In the ‘Bibliography’ six new References have been added.
10. In Sections II and III several mathematical expressions that previously appeared in the text have been put as equations.
11. In addition, minor changes have been made in several other places. Many of these are of grammatical nature.